# eIF2α-mediated translational control regulates the persistence of cocaine-induced LTP in midbrain dopamine neurons

Andon N Placzek[1,2†‡], Gonzalo Viana Di Prisco[1,2†], Sanjeev Khatiwada[1,2,3], Martina Sgritta[1,2], Wei Huang[1,2], Krešimir Krnjević[4], Randal J Kaufman[5], John A Dani[6], Peter Walter[7], Mauro Costa-Mattioli[1,2*]

[1]Department of Neuroscience, Baylor College of Medicine, Houston, United States; [2]Memory and Brain Research Center, Baylor College of Medicine, Houston, United States; [3]Verna and Marrs McLean Department of Biochemistry and Molecular Biology, Baylor College of Medicine, Houston, United States; [4]Department of Physiology, McGill University, Montreal, Canada; [5]Degenerative Diseases Program, SBP Medical Discovery Institute, La Jolla, United States; [6]Department of Neuroscience, Mahoney Institute for Neurosciences, Perelman School of Medicine, Philadelphia, United States; [7]Department of Biochemistry and Biophysics, Howard Hughes Medical Institute, University of California at San Francisco, San Francisco, United States

*For correspondence: costamat@bcm.edu

†These authors contributed equally to this work

Present address: ‡Division of Basic Medical Sciences, Mercer University School of Medicine, Macon, United States

Competing interests: The authors declare that no competing interests exist.

**Abstract** Recreational drug use leads to compulsive substance abuse in some individuals. Studies on animal models of drug addiction indicate that persistent long-term potentiation (LTP) of excitatory synaptic transmission onto ventral tegmental area (VTA) dopamine (DA) neurons is a critical component of sustained drug seeking. However, little is known about the mechanism regulating such long-lasting changes in synaptic strength. Previously, we identified that translational control by eIF2α phosphorylation (p-eIF2α) regulates cocaine-induced LTP in the VTA (Huang et al., 2016). Here we report that in mice with reduced p-eIF2α-mediated translation, cocaine induces persistent LTP in VTA DA neurons. Moreover, selectively inhibiting eIF2α-mediated translational control with a small molecule ISRIB, or knocking down *oligophrenin-1*—an mRNA whose translation is controlled by p-eIF2α—in the VTA also prolongs cocaine-induced LTP. This persistent LTP is mediated by the insertion of GluR2-lacking AMPARs. Collectively, our findings suggest that eIF2α-mediated translational control regulates the progression from transient to persistent cocaine-induced LTP.

## Introduction

Drug addiction is a complex behavioral disorder that starts with recreational use and, in some people, progresses to compulsive drug-seeking (*Hyman, 2005*). The precise molecular and cellular mechanism underlying this transition remains unclear. In addicts, repeated drug use leads to long-lasting changes in neuronal structure and function in key reward areas (*Koob and Volkow, 2010*), which have emerged as cellular correlates of drug addiction (*Chen et al., 2008*; *Lüscher and Malenka, 2011*). Of particular interest are excitatory synaptic afferents to dopaminergic neurons in the ventral tegmental area (VTA). These activity-dependent changes in synaptic strength in the VTA

constitute the initial synaptic adaptations observed after drug exposure (*Lüscher and Malenka, 2011*; *Schilstrom et al., 2006*). For instance, a single injection of cocaine (or other drugs of abuse) given 24 hr before recording induces an LTP that is manifested as an increase in the ratio of the amplitude of α-amino-3-hydroxy-5-methyl-4-isoxazolepropionic acid receptor (AMPAR)- to *N*-methyl D-aspartate receptor (NMDAR)-mediated excitatory postsynaptic currents (EPSCs) onto DA neurons in the VTA (*Ungless et al., 2001*). These early changes in synaptic strength are believed to facilitate persistent alterations in response to repeated exposure to drugs of abuse.

LTP induced by an acute injection of cocaine typically lasts up to five days, but it returns to baseline after 10 days (*Borgland et al., 2004*). This short-lasting LTP is associated with the insertion of AMPARs lacking the GluR2 subunit, as demonstrated by an increased inward rectification of postsynaptic AMPAR currents (*Bellone and Lüscher, 2006*; *Engblom et al., 2008*). Interestingly, in contrast to passive cocaine administration, LTP persists for several weeks following self-administration (*Chen et al., 2008*). Thus, these drug-induced persistent changes in synaptic strength in the VTA may represent the cellular process driving the progression from recreational to compulsive drug use. Furthermore, pharmacological activation of metabotropic glutamate receptors (mGluRs) with dihydroxyphenylglycine (DHPG), a group I mGluR agonist, inhibits cocaine-induced LTP in VTA DA neurons (*Bellone and Lüscher, 2006*). This inhibition is attributed to the opposing effect of mGluR-mediated long-term depression (mGluR-LTD), which reverses drug-evoked LTP and inward rectification in VTA DA neurons by internalization of AMPARs (*Bellone and Lüscher, 2006*; *Lüscher and Huber, 2010*; *Lüscher and Malenka, 2011*).

Our recent work provided a unifying model that explains how translational control by phosphorylation of the eukaryotic translation initiation factor eIF2 alpha subunit (eIF2α) regulates these two opposing forms of plasticity (cocaine-induced LTP and mGluR-induced LTD) in the VTA (*Huang et al., 2016*). First, cocaine induces LTP and reduces p-eIF2α levels in the VTA (*Huang et al., 2016*), whereas DHPG elicits mGluR1/5-induced LTD and increases p-eIF2α (*Di Prisco et al., 2014*; *Trinh et al., 2014*). Second, both genetic and pharmacological inhibition of p-eIF2α-mediated translational control facilitates the induction of LTP by blocking mGluR-LTD in VTA DA neurons (*Huang et al., 2016*). Finally, pharmacologically increasing p-eIF2α levels induces mGluR-LTD and prevents cocaine-induced LTP in VTA DA neurons (*Huang et al., 2016*).

Given that mGluR-LTD blocks the persistence of cocaine-induced LTP (*Mameli et al., 2009*), and our previous finding that p-eIF2α–mediated translational control regulates both forms of long-lasting plasticity (*Huang et al., 2016*), we examined whether the same translational control program regulates the shift from a relatively transient cocaine-induced LTP to a more persistent one in VTA DA neurons.

## Results

### Single or multiple injections of cocaine reduce eIF2α phosphorylation in the VTA

Single or repeated exposure to cocaine in rodents induces LTP that lasts for five days, but decays to baseline after 10 days (*Borgland et al., 2004*; *Kauer and Malenka, 2007*; *Mameli et al., 2009*; *Ungless et al., 2001*). To examine the effects of multiple doses of cocaine on p-eIF2α in the VTA, we injected mice with either saline or cocaine (10 mg/kg) once a day for five days and measured the phosphorylation levels of eIF2α in the VTA (*Figure 1a*). Interestingly, both acute and repeated exposure to cocaine reduced p-eIF2α levels in the VTA (*Figure 1b and c*).

### Cocaine elicits persistent LTP in mice with reduced eIF2α phosphorylation

We previously showed that the translational program controlled by p-eIF2α regulates the induction of LTP by a single injection of cocaine (*Huang et al., 2016*). To test whether the same process is involved in the persistence of LTP, we first employed *Eif2s1*[S/A] heterozygous mutant mice, in which a single phosphorylation site in eIF2α (encoded by the *Eif2s1* gene) at Serine-51 is replaced by alanine. These mice have reduced p-eIF2α levels in the VTA (*Huang et al., 2016*).

We injected control WT (*Eif2s1*[S/S]) and *Eif2s1*[S/A] mice with saline or cocaine (10 mg/kg) once a day for three days and recorded glutamate-mediated excitatory postsynaptic currents (EPSCs) from

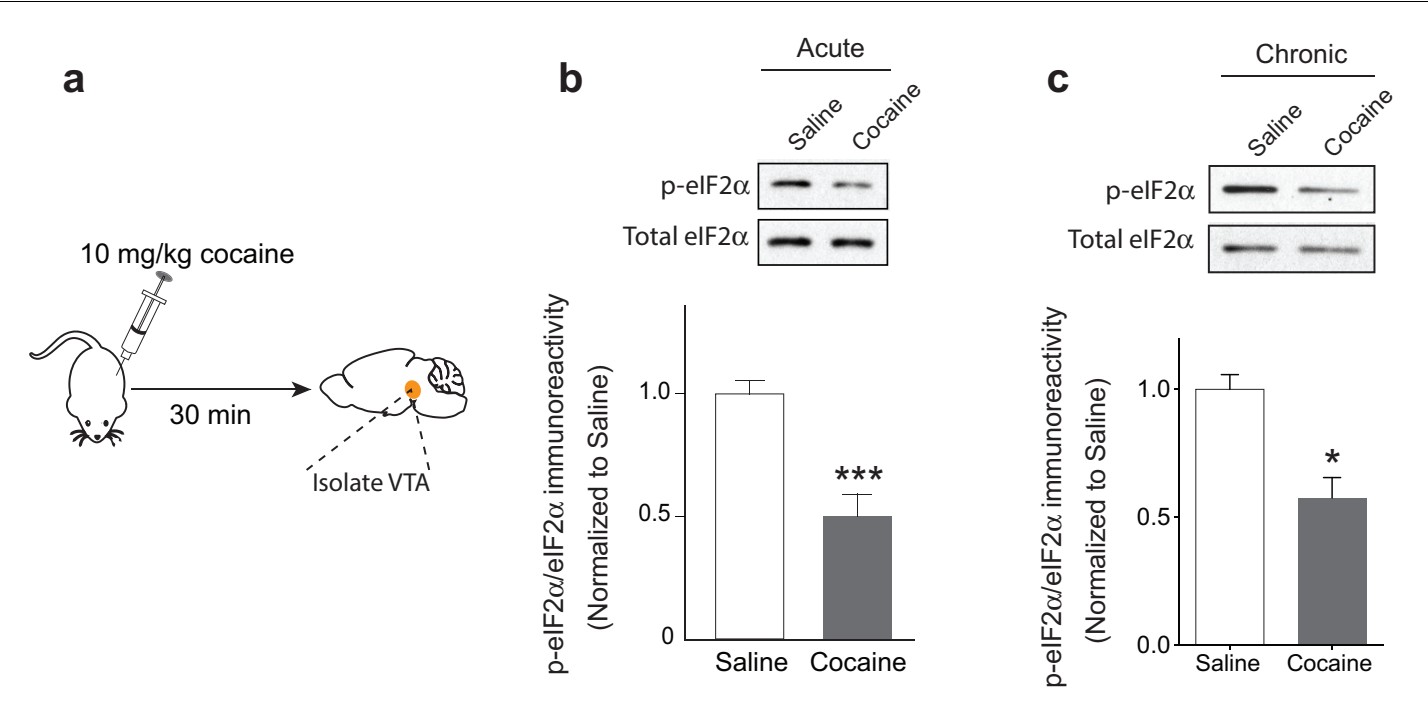

**Figure 1.** Acute and repeated exposure to cocaine reduces p-eIF2α levels in the VTA. (**a**) Schematic of experimental design. (**b-c**) Both single (10 mg/kg; p<0.001, $n = 6$ per group, $t_{10} = 4.640$) or multiple (five) injections of cocaine (10 mg/kg, p<0.05, $n = 3$ per group, $t_4 = 4.329$) reduced p-eIF2α levels in the VTA of adult mice.

VTA DA neurons in midbrain slices. We measured AMPAR/NMDAR ratios (recorded at +40 mV as we previously described (*Huang et al., 2016*; *Placzek et al., 2016*) 5 and 14 days after cocaine withdrawal. As previously reported (*Borgland et al., 2004*), in control slices, cocaine (10 mg/kg) elicited an LTP that lasted five days (5d) but not 14d after withdrawal (*Figure 2a and c*). By contrast, in VTA slices from *Eif2s1^{S/A}* mice, LTP was greater at five days compared to WT controls and was still present and not significantly changed 14 days after the last cocaine injection (*Figure 2b and c*). Indeed, cocaine-induced LTP in the VTA persisted for at least 40 days in *Eif2s1^{S/A}* mice, which was as long as we continued the withdrawal period (*Figure 2—figure supplement 1*). Thus, a decrease in eIF2α phosphorylation facilitates the progression from transient to persistent cocaine-induced LTP.

### Conditional reduction in eIF2α phosphorylation specifically in the VTA leads to persistent cocaine-induced LTP

To assess the regional and temporal specificity of our findings, we used a new eIF2α transgenic mouse line (*Eif2s1^{A/A}*;ftg) (*Di Prisco et al., 2014*) characterized by conditional expression of homozygous *Eif2s1* (Ser51Ala) mutants upon cell-type specific recombination mediated by Cre recombinase (*Back et al., 2009*). Cre-mediated deletion of the exogenous *Eif2s1* coding region coordinately induces the expression of green fluorescent protein (GFP) in a small population of VTA neurons, enabling us to manipulate p-eIF2α levels in single cells and record the consequences at the single-cell level (*Figure 2—figure supplement 2*). In midbrain slices from these mice, we recorded from GFP-positive (GFP+) neurons (in which eIF2α cannot be phosphorylated) and GFP-negative (GFP-) control neurons. As expected, in GFP- (control) VTA DA neurons, cocaine-evoked LTP lasted five days, but returned to baseline by 14 days after withdrawal (*Figure 2d and f*). However, in GFP+ VTA DA neurons lacking p-eIF2α, LTP was much greater at five days and persisted for at least 14 days (*Figure 2e and f*), highlighting the cell-autonomous effect of eIF2α phosphorylation on the persistence of LTP. It is noteworthy that AAV5-Cre-GFP could infect non-dopaminergic neurons in the VTA of *Eif2s1^{A/A}*;ftg mice. However, given the very low titer of virus injected to only sparsely infect

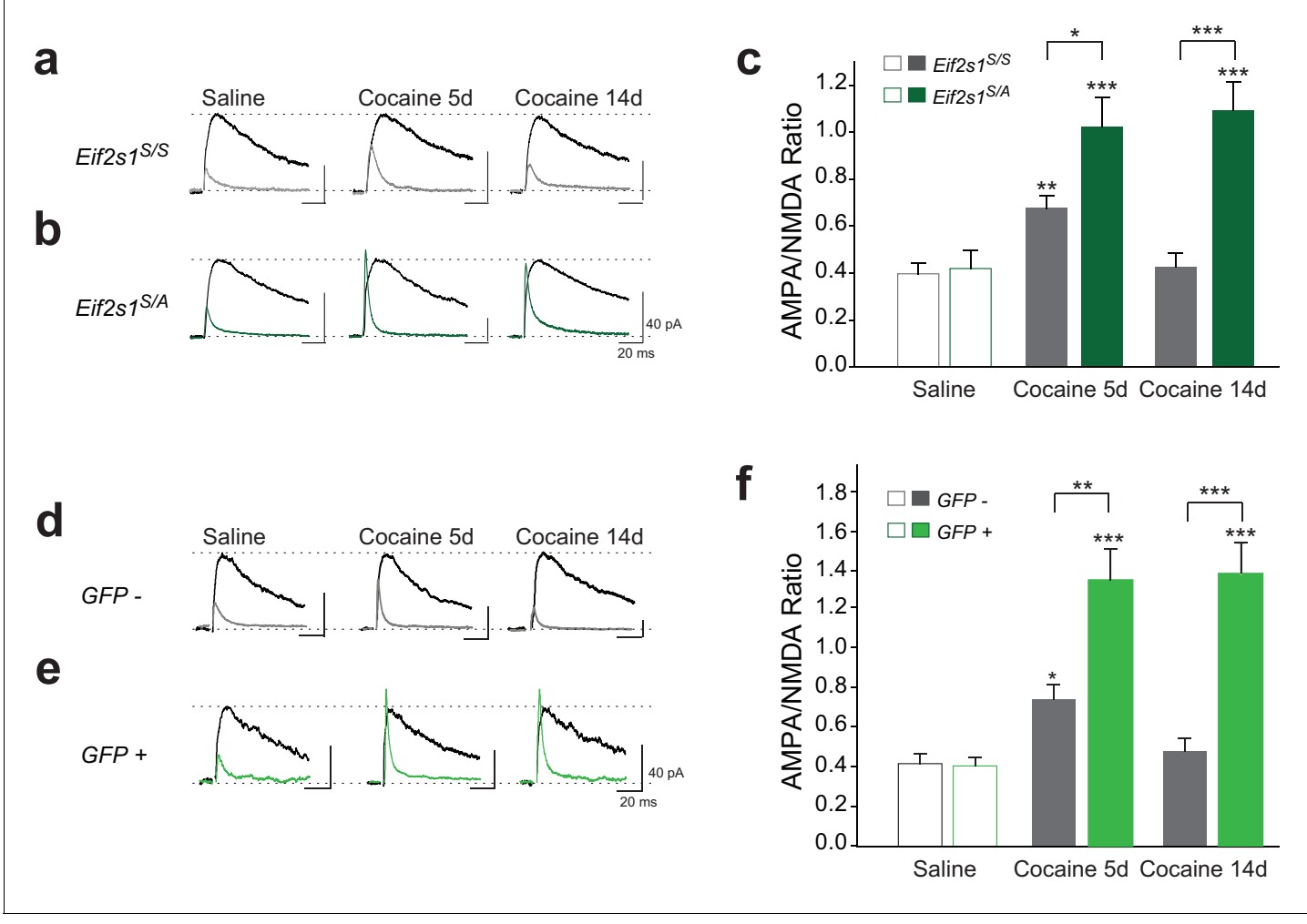

**Figure 2.** Reduced p-eIF2α levels render cocaine-induced LTP persistent in VTA DA neurons of adult mice. (a–c) Increased AMPAR/NMDAR ratios lasted only five days in cocaine-injected WT mice (10 mg/kg; 5d, p<0.01, $n$ = 11/9/10 saline/5d cocaine/14d cocaine, $F_{2,27}$ = 7.82), but persisted >14 days in cocaine-injected $Eif2s1^{S/A}$ mice (5d, p<0.001; 14d, p<0.001, $n$ = 9/8/10 saline/5d cocaine/14d cocaine, $F_{2,24}$ = 13.31). (d–f) Similarly, cocaine (10 mg/kg, i.p.) increased AMPAR/NMDAR ratio in GFP-positive cells (in which eIF2α cannot be phosphorylated), both at five and at 14 days post-injection (5d, p<0.001; 14d, p<0.001 , $n$ = 5/6/6 saline/5d cocaine/14d cocaine, $F_{2,14}$ = 8.30), compared to control GFP-negative cells (5d, p<0.05, $n$ = 5/8/5 saline/5d cocaine/14d cocaine, $F_{2,15}$ = 4.78) from $Eif2s1^{A/A}$;ftg mice.

The following figure supplements are available for figure 2:

**Figure supplement 1.** In $Eif2s1^{S/A}$ mice, cocaine induced LTP in VTA DA neurons persisted for 40 days.

**Figure supplement 2.** Selective absence of eIF2α phosphorylation in VTA enhances persistence of cocaine-induced LTP.

DA neurons in the VTA, we expect the effect of inputs from Cre-GFP expressing non-DA neurons to be minimal towards the persistence of LTP in GFP+ VTA DA neurons.

### Reduced p-eIF2α-mediated translational control leads to persistent cocaine-induced inward rectification

Cocaine-induced synaptic potentiation in VTA DA neurons is accompanied by a shift in the subunit composition of postsynaptic AMPARs (*Bellone and Lüscher, 2006*). Under normal conditions, post-synaptic AMPA-type currents are mediated by low conductance calcium-impermeable GluR1/GluR2 heteromeric receptors that are characterized by linear AMPAR current-voltage relationships. Exposure to cocaine results in the postsynaptic insertion of high conductance calcium-permeable AMPA

receptors that lack GluR2 subunits and are consequently inwardly rectifying (manifested as smaller AMPAR EPSCs measured at positive holding potentials versus those measured at corresponding negative potentials). In addition, GluR2-lacking AMPARs are sensitive to blockade by intracellular polyamines (*Iino et al., 1996*). We therefore sought to determine whether the persistent cocaine-induced LTP that we observed in *Eif2s1^{S/A}* mice was accompanied by a similarly long-lasting shift in AMPAR subunit composition. By recording EPSCs in VTA DA neurons at −70, 0, and +40 mV to calculate the rectification index, we found that cocaine increased inward rectification in control VTA DA neurons (*Eif2s1^{S/S}*; *Figures 3a,c and e*) and to a greater degree in slices from *Eif2s1^{S/A}* mice (*Figures 3b,d and e*) at five days post-injection. However, 14 days after cocaine treatment, EPSCs were non-rectifying in WT controls (*Figures 3a,c and e*), whereas *Eif2s1^{S/A}* mice retained their inwardly-rectifying properties (*Figures 3b,d and e*). Moreover, potentiated EPSCs in cocaine-treated *Eif2s1^{S/A}* mice measured at 14 days were sensitive to two blockers of GluR2-lacking AMPARs, Joro spider toxin (JST (*Iino et al., 1996*), 500 nM; *Figure 3—figure supplement 1a*) and 1-Naphthyl acetyl spermine (NASPM (*Tsubokawa et al., 1995*), 50 µM; *Figure 3—figure supplement 1b*). We conclude that postsynaptic insertion of GluR2-lacking AMPARs underlies the persistent LTP in VTA DA neurons with reduced p-eIF2α levels.

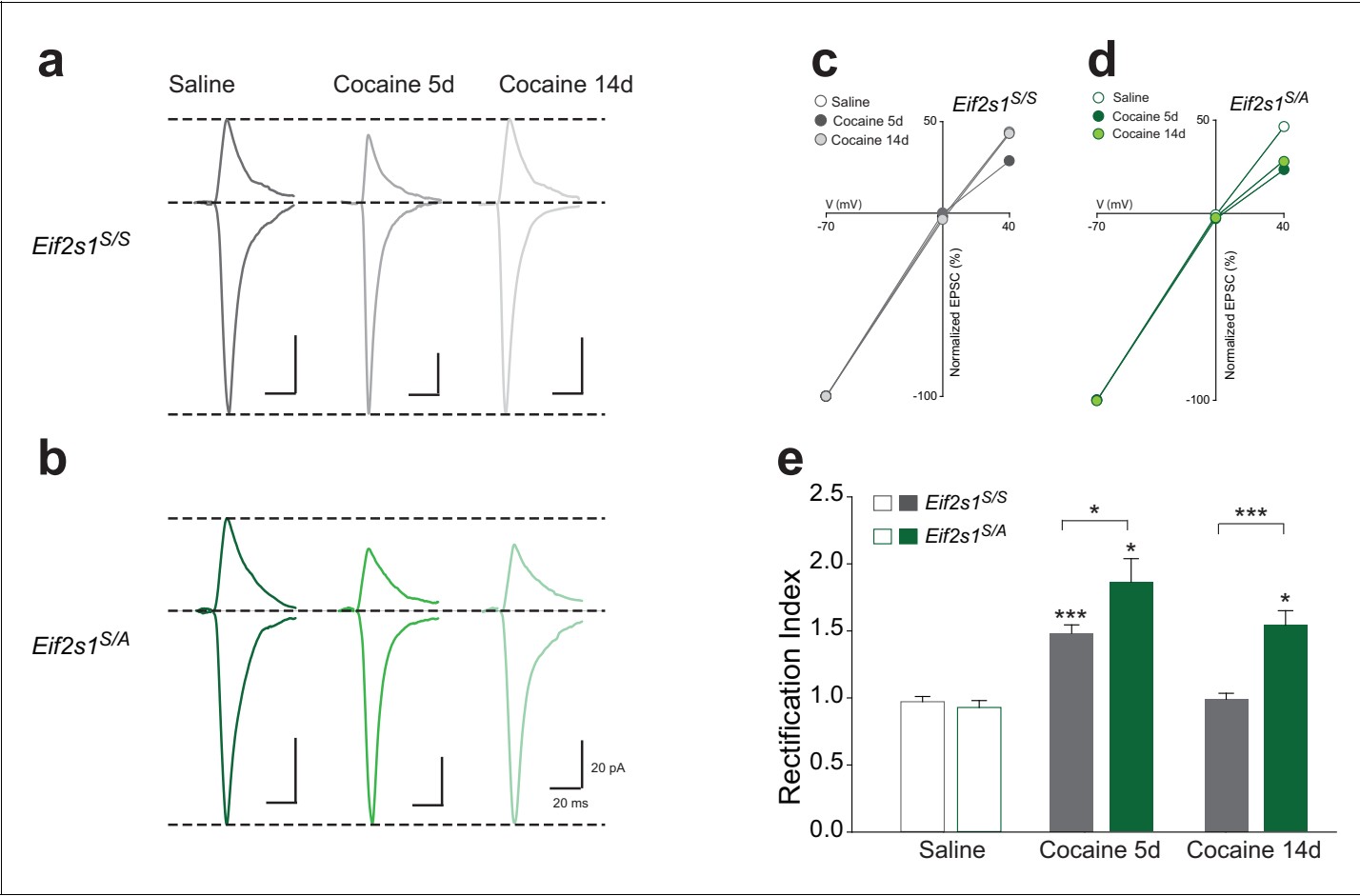

**Figure 3.** Reduction of eIF2α phosphorylation enhances the persistence of cocaine-induced AMPAR inward rectification. (a–b) Representative AMPAR EPSCs traces for each group recorded at −70 and +40 mV. *I-V* plots (c–d) and summaries (e) illustrate that cocaine-induced rectification lasts only five days in control mice (5d, $p < 0.001$, $n$ = 12/10/7 saline/5d cocaine/14d cocaine, $F_{2,26}$ = 32.02) but persists at least 14 days in *Eif2s1^{S/A}* mice (5d, $p < 0.05$; 14d, $p < 0.05$, $n$ = 8/8/10 saline/5d cocaine/14d cocaine, $F_{2,23}$ = 17.21).

The following figure supplement is available for figure 3:

**Figure supplement 1.** At 14 days post-injection, polyamines inhibit AMPAR EPSCs only in cocaine-treated *Eif2s1^{S/A}* mice.

# Selective pharmacological reduction of p-eIF2α-mediated translational control facilitates persistent cocaine-induced LTP and inward rectification

While phosphorylation of eIF2α inhibits general protein synthesis by inhibiting the guanine nucleotide exchange factor (GEF) eIF2B, it also increases the translation of specific mRNAs (*Buffington et al., 2014*; *Ron and Harding, 2007*). In our previous study, we used the small molecule inhibitor ISRIB, which promotes eIF2B activity resulting in selective reversal of p-eIF2α-mediated translational events (*Sekine et al., 2015*; *Sidrauski et al., 2013*, *2015*) and, consequently, makes adult mice more susceptible to the synaptic and behavioral effects of a sub-threshold dose of cocaine (*Huang et al., 2016*). To determine whether blocking p-eIF2α-mediated translation facilitates the switch from transient to persistent LTP, we administered WT mice with ISRIB (i.p. 2.5 mg/kg) followed by cocaine (10 mg/kg) once a day for three days. Consistent with our findings with *Eif2s1*$^{S/A}$ mice, cocaine induced both persistent LTP (*Figure 4a–c*) and inward rectification (*Figure 4d–h*) only in VTA DA neurons from ISRIB-injected mice.

Given that p-eIF2α-mediated synthesis of oligophrenin-1 (OPHN1) blocks cocaine-induced LTP (*Huang et al., 2016*), we next asked whether a reduction in OPHN1 levels in the VTA would also prolong cocaine-induced LTP. To answer this question, we knocked-down OPHN1 selectively in the VTA with a specific shRNA (*Ophn1*-shRNA) (*Nadif Kasri et al., 2011*). As expected, in OPHN1-deficient, but not in control VTA DA neurons, cocaine-induced LTP persisted for at least 14 days (*Figure 4i*). These data support the notion that cocaine-evoked LTP becomes persistent when p-eIF2α-mediated translation of OPHN1 is repressed.

## Discussion

Drug addiction is a disorder involving maladaptive plasticity in key reward areas of the brain (*Bowers et al., 2010*; *Hyman et al., 2006*; *Kauer and Malenka, 2007*; *Lüscher and Malenka, 2011*). In the VTA, the origin of the mesolimbic dopamine system, single or multiple injections of cocaine evoke synaptic potentiation that persists for about five days. This LTP is generated by the insertion of GluR2-lacking AMPARs into the postsynaptic membrane. Two studies revealed that mGluR-LTD in VTA DA neurons blocks cocaine-evoked LTP in these same cells by replacing GluR2-lacking calcium permeable AMPARs with GluR2-containing receptors (*Bellone and Lüscher, 2006*; *Mameli et al., 2007*). Unlike NMDA-induced LTD, mGluR-LTD depends on new protein synthesis (*Lüscher and Huber, 2010*). We have previously shown that eIF2α-mediated translational control is necessary and sufficient for mGluR-LTD in both the hippocampus and the VTA (*Di Prisco et al., 2014*; *Huang et al., 2016*). Translational control by the mammalian target of rapamycin complex 1 (i.e., mTORC1) (*Buffington et al., 2014*; *Costa-Mattioli and Monteggia, 2013*) has also been implicated in mGluR-LTD. Treatment with the mTORC1 inhibitor rapamycin blocks mGluR-LTD in the VTA (*Mameli et al., 2007*). While both translational control mechanisms (mTORC1 and p-eIF2α) could be required for mGluR-LTD, it is possible that rapamycin treatment could affect eIF2α-mediated translational control. During mGluR-LTD, activation of mGluRs by DHPG triggers the synthesis of OPHN1 (*Nadif Kasri et al., 2011*) in a p-eIF2α-dependent manner (*Di Prisco et al., 2014*). If rapamycin exerted its effects on mGluR-LTD by blocking OPHN1 translation, which is required to induce mGluR-LTD (*Di Prisco et al., 2014*; *Nadif Kasri et al., 2011*), these seemingly contradictory findings would be easily reconciled.

Inhibition of mGluR-LTD by genetic disruption of Homer1C-mGluR1 interaction or by treatment with an antagonist of group I mGluRs [1-aminoindan-1,5-dicarboxylic acid (AIDA)] has been shown to prolong cocaine-induced LTP from five days to seven days (*Mameli et al., 2009*). By suppressing p-eIF2α-mediated mGluR-LTD, we found that cocaine induces an LTP that lasted up to 40 days and was mediated by the insertion of GluR2-lacking receptors. Thus, like self-administration of cocaine (*Chen et al., 2008*), passive injections of cocaine in mice with reduced p-eIF2α-mediated translational control elicit a synaptic potentiation that lasts for weeks.

Recreational drug use is clinically separable from escalated drug use and compulsive drug-seeking behavior that characterize addiction. Our findings indicate that p-eIF2α-mediated translation prevents the progression from transient to persistent cocaine-induced LTP in VTA DA neurons, thus regulating synaptic potentiation that is believed to underlie compulsive drug seeking (*Chen et al., 2008*; *Lüscher and Malenka, 2011*; *Mameli et al., 2009*). In future studies, it will be interesting to

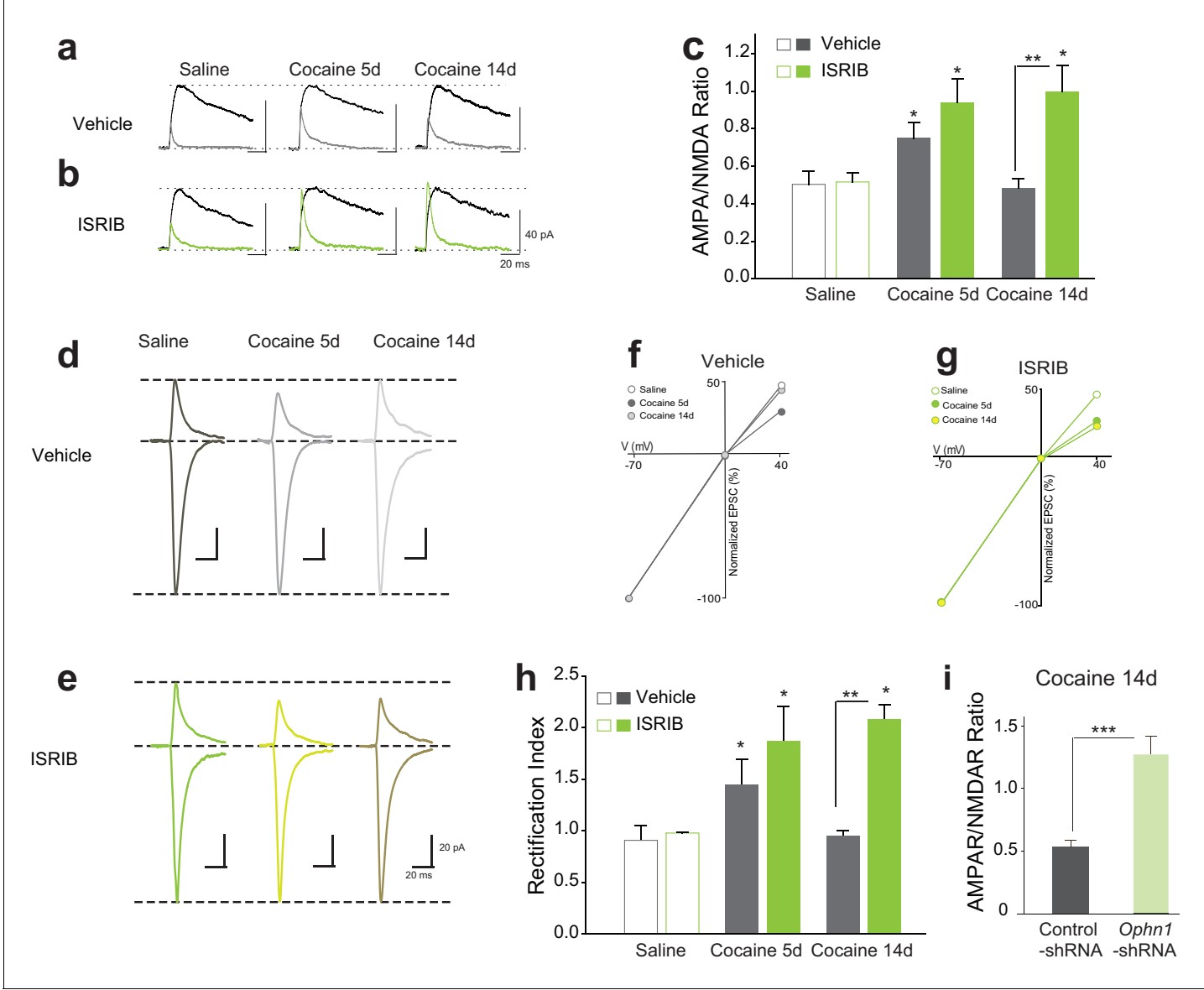

**Figure 4.** Reduced p-eIF2α-mediated translation enhances the persistence of cocaine-induced LTP and rectification in VTA DA neurons. (**a–c**) Cocaine-induced increase in AMPAR/NMDAR ratios lasted only five days in control mice (5d, p<0.05, $n$ = 5/5/5 saline/5d cocaine/14d cocaine, $F_{2,12}$ = 8.38), but persisted at least 14 days in ISRIB-injected mice (5d, p<0.05; 14d, p<0.05, $n$ = 5/5/5 saline/5d cocaine/14d cocaine, $F_{2,12}$ = 6.40). (**d–e**) Representative AMPAR EPSCs traces and AMPAR/NMDAR ratios. *I-V* plots (**f–g**) and summaries (**h**) illustrate that cocaine-induced rectification lasts only five days in vehicle-injected mice (5d, p<0.05, $n$ = 4/4/4 saline/5d cocaine/14d cocaine, $F_{2,9}$ = 7.12), but persists at least 14 days in ISRIB-injected mice (5d, p<0.05; 14d, p<0.05, $n$ = 4/4/4 saline/5d cocaine/14d cocaine, $F_{2,9}$ = 7.53). (**i**) Knocking down OPHN1 in the VTA prolonged cocaine-induced LTP to 14 days after cocaine withdrawal (p<0.001, $n$ = 9/8 control-shRNA/*Ophn1*-shRNA, $t_{15}$ = 4.986).

examine whether this translational control mechanism regulates drug-seeking behaviors. In addition, it will important to assess whether cocaine-induced LTP could prime mice for enhanced behavioral output when challenged with a low dose of cocaine (or other drug of abuse) within the persistence window. Finally, it will be intriguing to study whether blocking p-eIF2α-mediated translation with ISRIB during extinction of cocaine memory is sufficient to prevent reinstatement.

In humans, the vulnerability to compulsive drug abuse has been linked to deficits in cortico-striatal processing (*Ersche et al., 2011*; *Volkow et al., 2009*). We have recently found that mesolimbic neuronal reward responses are altered in human smokers carrying a single nucleotide polymorphism in

the *Eif2s1* gene (which encodes eIF2α) (*Placzek et al., 2016*). Therefore, it would be interesting to examine whether cortico-striatal processing is also dysfunctional in these individuals. In this regard, we speculate that in other key reward areas, mice with reduced p-eIF2α-mediated translational control should exhibit synaptic adaptations associated with persistent LTP in the VTA (*Mameli et al., 2009*).

Finally, in cocaine addicts, persistent activity-dependent changes in synaptic strength in the VTA are likely to produce long-lasting changes in synaptic function in downstream structures, such as the nucleus accumbens (NAc) and prefrontal cortex (PFC). Thus, for a treatment for addiction to be effective, ideally, it should not only target the initial synaptic neuroadaptations in the VTA, but also the synaptic function in other related structures. Given that enhancing mGluR signaling in the NAc blocks cocaine relapse and craving (*Loweth et al., 2013*) and that eIF2α is a key regulator of mGluR-LTD and cocaine-induced LTP, we propose that modulators of p-eIF2α-mediated translational control may be useful in the treatment of cocaine addiction.

# Materials and methods

## Mice

All experiments were conducted using C57Bl/6J male and female mice. *Eif2s1^{S/A} and Eif2s1^{A/A}*;ftg mice were previously described (*Di Prisco et al., 2014*). Mice were kept on a 12 hr/12 hr light/dark cycle (lights on at 7:00 am) and had access to food and water *ad libitum*. Animal care and experimental procedures were approved by the institutional animal care and use committee (IACUC) at Baylor College of Medicine, according to NIH Guidelines.

No statistical methods were used to predetermine sample sizes. All sample sizes meet the criteria for corresponding statistical tests—our sample sizes are similar to those reported in previous publications (*Argilli et al., 2008*; *Bellone and Lüscher, 2006*; *Koo et al., 2012*; *Saal et al., 2003*; *Ungless et al., 2001*).

## Drug treatment

Cocaine was dissolved in 0.9% saline and injected at a volume of 5 ml/kg. Cocaine hydrochloride was obtained from Sigma-Aldrich (St. Louis, MO). ISRIB (P. Walter) was dissolved in DMSO and further diluted in PEG-400 (1:1 ratio) as previously described (*Sidrauski et al., 2013*). For both electrophysiological and behavioral experiments, ISRIB (2.5 mg/kg) or vehicle (DMSO/PEG-400, 2 ml/kg) was injected 90 min before cocaine or saline injection.

## Slice electrophysiology

Electrophysiological recordings were performed as previously described (*Huang et al., 2016*; *Placzek et al., 2016*). The investigators were kept blind to genotypes, and each electrophysiology experiment was replicated at least three times. Briefly, mice were anesthetized with a mixture of ketamine (100 mg/kg), xylazine (10 mg/kg), and acepromazine (3 mg/kg). Mice were transcardially perfused with an ice-cold, oxygenated (95% $O_2$/5% $CO_2$) solution containing (in mM) NaCl, 120; $NaHCO_3$, 25; KCl, 3.3; $NaH_2PO_4$, 1.2; $MgCl_2$, 4; $CaCl_2$, 1; dextrose, 10; sucrose, 20. Horizontal slices (225–300 μm thick) containing the VTA were cut from the brains of adult (3–5 months old) C57BL/6J mice with a vibrating tissue slicer (VF-100 Compresstome, Precisionary Instruments, San Jose, CA, or Leica VT 1000S, Leica Microsystems, Buffalo Grove, IL). Slices were incubated at 34°C for 40 min, at room temperature for at least 30 min prior to recording, and then transferred to a recording chamber and continuously perfused with oxygenated artificial cerebrospinal fluid (ACSF) at 32°C and a flow rate of 2–3 ml/min. The recording ACSF was identical to the cutting solution except for the concentration of $MgCl_2$ (1 mM) and $CaCl_2$ (2 mM). Recording pipettes were made from thin-walled borosilicate glass (TW150F-4, WPI, Sarasota, FL), which were filled with intracellular solution containing (in mM): 117 CsMeSO3; 0.4 EGTA; 20 HEPES; 2.8 NaCl, 2.5 ATP-Mg 2.0; 0.25 GTP-Na; 5 TEA-Cl, adjusted to pH 7.3 with CsOH and 290 mOsmol/l; tip resistance, 3–5 MΩ. For studies of AMPAR current rectification, spermine (100 μM) was added to the internal solution.

Data were obtained with a MultiClamp 700B amplifier, digitized at 20 kHz with a Digidata 1440A, recorded by Clampex 10 and analyzed with Clampfit 10 software (Molecular Devices), and filtered online at 4 kHz with a low-pass Bessel filter. A 2 mV hyperpolarizing pulse was applied before each

EPSC to evaluate the input ($R_i$)and access resistances ($R_a$). Data were not included if $R_a$ was either unstable or greater than 25 MΩ, holding current was >200 pA, $R_i$ dropped >20% during the recording, or EPSC baselines changed by >10%. The representative traces illustrated in Figures are averages of 10–15 consecutive recorded sweeps.

After establishing a gigaohm seal (>2 GΩ) and recording stable spontaneous firing in cell-attached, voltage clamp mode (−70 mV holding potential), cell phenotype was determined by measuring the width (>1.0 ms) of the cell-attached action potential (*Chieng et al., 2011*; *Ford et al., 2006*). AMPAR/NMDAR ratios were calculated as previously described (*Ungless et al., 2001*). Briefly, neurons were voltage-clamped at +40 mV until the holding current stabilized (at <200 pA), which usually occurred after a period of 5 to 15 min. A bipolar stimulating electrode placed 50–150 μm rostral to the lateral VTA was used to evoke stable monosynaptic EPSCs at 0.05 Hz. Picrotoxin (100 μM) was added to the recording ACSF to block GABA$_A$R-mediated IPSCs. After recording the dual-component EPSC, bath-application of DL-AP5 (100 μM for 10 min) was used to remove the NMDAR component, which was then obtained by offline subtraction of the remaining AMPAR component from the original dual-component EPSC. The peak amplitudes of the isolated currents were used to calculate the AMPAR/NMDAR ratios. Rectification indices were calculated as the ratio of the chord conductance of evoked EPSCs at a negative holding potential (−70 mV) to the chord conductance obtained from recordings made at a positive holding potential (+40 mV) in the presence of 100 μM DL-AP5, as previously described (*Bellone and Lüscher, 2006*). Joro spider toxin (JST) was obtained from Sigma-Aldrich (St. Louis, MO), Picrotoxin and DL-AP5 were purchased from Tocris Bioscience, and all other reagents and experimental compounds were obtained from Sigma-Aldrich.

### Virus injection

AAV5-Cre (Titer: 1.0e13GC/ml) was purchased from Vector Biolabs (Cat#7012, Philadelphia, PA); Lentiviral constructs expressing *Ophn1* shRNA and scrambled shRNA were generously provided by Dr. Linda van Aelst (*Nadif Kasri et al., 2011*) and viruses were produced by Gene Vector Core Laboratory (Baylor College of Medicine). Viral injections were performed as previously described (*Di Prisco et al., 2014*). Briefly, mice were anaesthetized with isoflurane (2–3%) and viruses (1–2 μl/site) were injected bilaterally at the rate of 0.1 μl/min, and an additional 10 min to allow for diffusion of viral particles. Injection coordinates targeting the VTA were as follows (with reference to bregma): −2.50 AP, ± 0.45 ML, −4.50 DV. The incision was sutured after injection and mice were returned to home cages. Mouse body weight and signs of illness were monitored until full recovery from surgery (~1 week). Drug treatment and experiments were all performed at least three weeks after viral injection.

### Statistical analyses

All data are presented as mean ± s.e.m. Statistical analyses were performed using SigmaPlot (Systat Software). Data distribution normality and homogeneity of variance were assessed using the Shapiro-Wilk and Levene tests, respectively. The statistics were based on the two-sided Student's t test, or one- or two-way ANOVA with Tukey's HSD (or HSD for unequal sample sizes where appropriate) to correct for multiple *post hoc* comparisons. Within-groups variation is indicated using standard errors of the mean of each distribution, which are depicted in the graphs as error bars. $p < 0.05$ was considered significant (*$p < 0.05$, **$p < 0.01$, ***$p < 0.001$, ****$p < 0.0001$).

## Acknowledgements

We thank Hongyi Zhou for assisting in the maintenance of mouse colony and members of the Costa-Mattioli laboratory for comments on the paper. This work was supported by grants from the National Institutes of Health (NIMH 096816, NINDS 076708) and the generous support from Sammons Enterprises. RJK is supported by NIH grants DK042394, DK088227, DK103183 and CA128814. JAD was supported by grants NIH NIDA DA09411 and NINDS NS21229. PW is an investigator of the Howard Hughes Medical Institute.

## Additional information

### Funding

| Funder | Grant reference number | Author |
|---|---|---|
| National Institute of Diabetes and Digestive and Kidney Diseases | DK042394 | Randal J Kaufman |
| National Institute of Diabetes and Digestive and Kidney Diseases | DK088227 | Randal J Kaufman |
| National Institute of Diabetes and Digestive and Kidney Diseases | DK103183 | Randal J Kaufman |
| National Cancer Institute | CA128814 | Randal J Kaufman |
| National Institute on Drug Abuse | DA09411 | John A Dani |
| National Institute of Neurological Disorders and Stroke | NS21229 | John A Dani |
| Howard Hughes Medical Institute | | Peter Walter |
| National Institute of Mental Health | MH09816 | Mauro Costa-Mattioli |
| National Institute of Neurological Disorders and Stroke | NS076708 | Mauro Costa-Mattioli |

The funders had no role in study design, data collection and interpretation, or the decision to submit the work for publication.

### Author contributions

ANP, Conception and design, Acquisition of data, Analysis and interpretation of data, Drafting or revising the article; GVDP, Conception and design, Acquisition of data, Analysis and interpretation of data; SK, Acquisition of data, Analysis and interpretation of data, Drafting or revising the article; MS, Acquisition of data, Analysis and interpretation of data; WH, Conception and design, Acquisition of data; KK, Drafting or revising the article; RJK, Drafting or revising the article, Contributed unpublished essential data or reagents; JAD, PW, Conception and design, Drafting or revising the article; MC-M, Conception and design, Analysis and interpretation of data, Drafting or revising the article

### Ethics

Animal experimentation: Animal care and experimental procedures were approved by the institutional animal care and use committee (IACUC) at Baylor College of Medicine (Protocol number: AN-5068), according to NIH Guidelines.

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
