## [Decision Letter]

Thank you for submitting your article "eIF2α-mediated translational control regulates the persistence of cocaine-induced LTP in midbrain dopamine neurons" for consideration by *eLife*. Your article has been favorably evaluated by a Senior Editor and two reviewers, one of whom, Rui M Costa (Reviewer #2), is a member of our Board of Reviewing Editors.

The reviewers have discussed the reviews with one another and the Reviewing Editor has drafted this decision to help you prepare a revised submission.

The study is seen as very interesting, but there are a few points to check before it can be considered further.

The authors should clarify the effects on total eIF2α p-eIF2α levels, provide evidence or discuss the specificity of their manipulation, and check if similar effects are observed with another drug (NASPM is suggested).

A direct link between eIF2α and OPHN, or a correlating between LTP at 14 days and behavior would be great additions, but are long experiments and so if the authors already have some data they could add it, otherwise these aspects could be touched upon in the Discussion.

Below are the specific comments to help prepared the revised submission.

*Reviewer #1:*

This is a very interesting study, also perhaps a little bit on the brief side for *eLife*. The findings seem to support the conclusions, but the manuscript would benefit a lot from a direct link of the physiology to cocaine's behavioral effects, even just simple locomotor activity or sensitization.

For Figure 1, was there a significant difference in the total eIF2α and p-eIF2α levels, or was the effect only significant for the ratio of p-eIF2α/total eIF2α?

For the AAV5-Cre-GFP virus injection into the VTA of eIF2α floxed mice, it is not clear whether the effect is mediated by selective knockdown of expression in VTA neurons, let alone DA neurons, as AAV5-Cre has strong retrograde activity. The authors should consider a way to knock it down from DA neurons selectively (i.e. crossing to DAT-Cre mouse) or state the caveat of the AAV5-Cre approach as non-specific to VTA.

JST is not the canonical drug utilized to block GluA2-lacking AMPA receptors. It would be important to see if similar effects are observed with NASPM.

A direct link between eIF2α and OPHN1 would make the study's final conclusion much more convincing. For example, overexpressing OPHN1 in eIF2α mutants should rescue heightened, long-lasting LTP.

*Reviewer #2:*

This Research Advance builds on previous findings by the same group, and helps elucidate the role of eIF2α in cocaine-induced plasticity in the ventral tegmental area. The authors find that single or repeated injections of cocaine lead to reduced levels of eIF2α phosphorylation in the ventral tegmental area. Furthermore they confirm that cocaine administration elicits long-term potentiation of glutamatergic synapses onto ventral tegmental area dopaminergic neurons that lasts at least 5 days but decays to baseline after 14 days. However, in mice with reduced eIF2α phosphorylation the potentiation is long-lasting and persist even after 14 days.

The authors then show that selective pharmacological reduction of eIF2α mediated translation has the same has the same effect and facilitates persistent cocaine-induced LTP even after 14 days. The authors have previously shown that eIF2α-mediated synthesis of OPHN1 blocks cocaine-induced LTP. So finally, here they confirmed that cocaine-induced LTP persisted for 14 days in OPHN1-deficient, but not in control VTA DA neurons.

These findings add mechanistic insight to the previous findings, are very well done and lead to straightforward conclusions. The only thing to add (if the authors would already have it) would be a behavioral correlate of the 14 days persistent LTP in the conditions applied here (before animals in the different groups were sacrificed).

---

## [Author Response]

*[…] The study is seen as very interesting, but there are a few points to check before it can be considered further.*

*The authors should clarify the effects on total eIF2α p-eIF2α levels, provide evidence or discuss the specificity of their manipulation, and check if similar effects are observed with another drug (NASPM is suggested).*

*A direct link between eIF2α and OPHN, or a correlating between LTP at 14 days and behavior would be great additions, but are long experiments and so if the authors already have some data they could add it, otherwise these aspects could be touched upon in the Discussion.*

To our knowledge, the best behavioral correlate of persistent LTP of excitatory synaptic transmission onto VTA DA neurons is compulsive drug-seeking behavior (see Mameli et al., Nat Neurosci, 2009; Chen et al., Neuron, 2008). Unfortunately, the self-administration procedure in mice is very technically challenging and would take well beyond the two months allowed by the journal for resubmission. While interesting, we believe that the study of whether p-eIF2α -mediated translational control regulates compulsive drug-seeking behavior is beyond the scope of this study. That said, we now address this issue in the Discussion section (third paragraph). In future studies, we plan to assess whether reduced p-eIF2α -mediated translation in the VTA precipitates compulsive cocaine seeking.

*Below are the specific comments to help prepared the revised submission.*

*Reviewer #1:*

*This is a very interesting study, also perhaps a little bit on the brief side for eLife. The findings seem to support the conclusions, but the manuscript would benefit a lot from a direct link of the physiology to cocaine's behavioral effects, even just simple locomotor activity or sensitization.*

*For Figure 1, was there a significant difference in the total eIF2α and p-eIF2α levels, or was the effect only significant for the ratio of p-eIF2α/total eIF2α?*

We examined the effect of chronic cocaine treatment on both p-eIF2α and total eIF2α levels and found that only the levels of phosphorylated eIF2α significantly change in response to cocaine. New representative figures are now provided (Figure 1).

*For the AAV5-Cre-GFP virus injection into the VTA of eIF2α floxed mice, it is not clear whether the effect is mediated by selective knockdown of expression in VTA neurons, let alone DA neurons, as AAV5-Cre has strong retrograde activity. The authors should consider a way to knock it down from DA neurons selectively (i.e. crossing to DAT-Cre mouse) or state the caveat of the AAV5-Cre approach as non-specific to VTA.*

As suggested by the reviewer, we now discussed the technical limitation of using AAV5-Cre to excise the floxed WT eIF2α transgene in the VTA (subsection “Conditional reduction in eIF2α phosphorylation specifically in the VTA leads to persistent cocaine-induced LTP”).

*JST is not the canonical drug utilized to block GluA2-lacking AMPA receptors. It would be important to see if similar effects are observed with NASPM.*

We agree with the reviewer’s comment. As suggested, we have now performed a new set of experiments in which we examined the effect of NASPM on AMPAR- mediated EPSCs in slices from *Eif2s1^S/S^*and *Eif2s1^S/A^*mice. Consistent with the JST results, NASPM blocked AMPAR-mediated EPSC in slices from *Eif2s1^S/A^*mice, but not in slices from *Eif2s1^S/S^*mice (see new Figure 3—figure supplement 1). Thus, these data further support the conclusion that the persistent LTP in *Eif2s1^S/A^*mice is due to the insertion of GluR2-lacking AMPARs on the synaptic surface.

*A direct link between eIF2α and OPHN1 would make the study's final conclusion much more convincing. For example, overexpressing OPHN1 in eIF2α mutants should rescue heightened, long-lasting LTP.*

In our previous paper (Huang et al., *eLife,*2016), we provided causal evidence that the effects of eIF2α phosphorylation on cocaine-induced LTP are mediated, at least in part, by OPHN1. We showed that Sal003, which promotes phosphorylation of eIF2α, blocks cocaine-induced LTP in control neurons, but not in neurons in which OPHN1 levels are reduced.

While we agree that overexpressing OPHN1 should prevent the persistence of LTP in *Eif2s1^S/A^*mice, we are concerned that overexpressing OPHN1 might lead to changes in basal synaptic transmission as has been previously reported (Kasri et al., JN, 2009) and might occlude cocaine-induced LTP.